# Adaptive Choropleth Mapper: An Open-Source Web-Based Tool for Synchronous Exploration of Multiple Variables at Multiple Spatial Extents

**Su Yeon Han [1],\*, Sergio Rey [1], Elijah Knaap [1], Wei Kang [1]  and Levi Wolf [2]**

[1]  Center for Geospatial Sciences, University of California Riverside, Riverside, CA 92521, USA;
    sergio.rey@ucr.edu (S.R.); knaap@ucr.edu (E.K.); weikang@ucr.edu (W.K.)
[2]  School of Geographical Sciences, University of Bristol, Bristol BS8 1SS, UK; levi.john.wolf@bristol.ac.uk
\*  Correspondence: suhan@ucr.edu

**Abstract:** Choropleth mapping is an essential visualization technique for exploratory spatial data analysis. Visualizing multiple choropleth maps is a technique that spatial analysts use to reveal spatiotemporal patterns of one variable or to compare the geographical distributions of multiple variables. Critical features for effective exploration of multiple choropleth maps are (1) automated computation of the same class intervals for shading different choropleth maps, (2) dynamic visualization of local variation in a variable, and (3) linking for synchronous exploration of multiple choropleth maps. Since the 1990s, these features have been developed and are now included in many commercial geographic information system (GIS) software packages. However, many choropleth mapping tools include only one or two of the three features described above. On the other hand, freely available mapping tools that support side-by-side multiple choropleth map visualizations are usually desktop software only. As a result, most existing tools supporting multiple choropleth-map visualizations cannot be easily integrated with Web-based and open-source data visualization libraries, which have become mainstream in visual analytics and geovisualization. To fill this gap, we introduce an open-source Web-based choropleth mapping tool called the Adaptive Choropleth Mapper (ACM), which combines the three critical features for flexible choropleth mapping.

**Keywords:** open-source GIS; choropleth map; Web-based GIS; geovisualization

---

## 1. Introduction

A choropleth map visualizes different values of measurement or a property with different shading or coloring across different geographic regions [1]. Choropleth maps visualize ordinal or interval/ratio data. To visualize data in a choropleth map, class intervals are computed according to a classification scheme, and then a graduated color palette is used to differentiate polygons belonging to each class. The purpose of a classification scheme is to group similar values into the same class and separate values that are different into different classes. Over the years, a number of different algorithms have been proposed, and each can result in a different visual pattern in the choropleth map. Mapmakers typically must choose the best classification algorithm for a particular map.

Choropleth maps have been used extensively to visualize the spatial characteristics of data on disaster-damaged areas [2–5], crime events [6,7], health outcomes [8–10], mortality rates and causes of death [11], and social media messages and Web content [12–14]. Also, the US Census Bureau, National Center for Health Statistics and National Cancer Institute produce many choropleth maps for researchers, policymakers, and educators [15–17]. Simply put, choropleth maps are a ubiquitous method for visualizing spatial data.

To reveal the spatiotemporal patterns of a particular variable (e.g., Figure S1) or to compare the spatial distributions of different variables within the same region (e.g., Figure S2), it is often necessary to create multiple choropleth maps. In this case, each map represents a different variable or time-step. In order for these multiple choropleth maps to be effective, there are three critical components: (1) an automated method to create multiple choropleth maps with identical class intervals, (2) a function that allows users to promptly recreate class intervals within any zoomed-in area, and (3) linking and brushing to enable synchronous exploration of multiple choropleth maps. The first component enables a direct comparison between the visual patterns in multiple maps, the so-called small multiples. The second component recreates class intervals within focused regions, automatically avoiding repetitive operations in a user interface. The third component allows users to zoom in/out or pan multiple maps together so that all maps always display the same region.

The extensive use of choropleth maps has led many GIS software packages to include a choropleth mapping tool, and several software packages support multiple choropleth map visualizations. However, existing GIS software packages include only one or two of the three components mentioned above. Also, many of them are desktop software, or they are commercial GIS software packages. Thus, they are not easily extensible with open-source JavaScript libraries, which have gained popularity in visual analytics and geovisualization (see Section 2.2. for details).

In terms of the three critical components for effective multiple choropleth maps mentioned above, the second component is rare in modern GIS software packages, while the first and the third components are common. Indeed, the lack of local analytical visualization tools in modern GIS software packages is surprising given the prevalence of local statistical techniques. In the spatial statistics literature, there is a long history of developing local forms of spatial statistical methods. For example, the Local Indicators of Spatial Association (LISA) statistic and associated visualizations [18] provide a general framework to decompose global spatial autocorrelation statistics, including the well-known Moran's I [19] and Geary's C [20]. LISA statistics can be used to reveal not only the instability of spatial autocorrelation but also to identify hot and cold spots which deserve further investigation. However, when we visually explore detailed spatial patterns within these hot spots using choropleth mapping methods in popular modern software packages such as ArcGIS Pro or Tableau, we may fail to uncover enough details. To reveal enough detail, we would need to create another choropleth map that focuses on a region of the map that includes the selected data in each hot spot or cold spot. In other words, users must subset and reconstruct the input datasets, treating the new view as an entirely different dataset. Since users usually recreate visualizations many times with different variables over many different subregions, and at many different spatial extents, the user experience is repetitive and error-prone. While conceptually simple and analytically attractive, the labor-intensive work makes it harder for spatial scientists to get a feel for their data.

In summary, although innovative features for a dynamic exploration of multiple choropleth maps have been developed, no existing choropleth mapping tool provides all the three critical components mentioned above in one open-source software package. Furthermore, many of them are difficult to integrate with popular Web-centered JavaScript data visualization libraries. To fill the gap in choropleth mapping implementations, we developed an open-source Web-based choropleth mapping tool called the Adaptive Choropleth Mapper (ACM), which integrates the three critical components mentioned above. Using the ACM, users can (1) automatically compute and set the same class intervals for one or more choropleth maps, (2) visualize the local spatial structure of choropleth maps easily with the click of one button, and (3) link multiple choropleth maps, with each representing a different variable, for a side-by-side comparison. We also suggest a Web-based GIS application called Longitudinal Neighborhood Explorer (LNE), which embeds ACM and offers tailored spatiotemporal data analysis.

The rest of the paper is organized as follows. The problem section identifies the limitations of currently available choropleth mapping tools in both desktop-based and Web-based geographic information systems, especially in terms of Web-based GIS applications visualizing spatiotemporal census or American Community Survey (ACS) data. We then discuss the Adaptive Choropleth Mapper

and how it liberates conventional choropleth mapping from global frames of analysis. The solution section first describes the motivation of this study, the data used, a new design of the user interface to visualize spatiotemporal data or multiple variables, and a suite of functionalities in the ACM that distinguishes it from conventional choropleth mapping practices. Finally, the conclusion section summarizes this study and mentions where users can access the ACM, which is provided as free and open-source software to support open science.

## 2. The Problem

### 2.1. Limitations of Choropleth Mapping Tools in Currently Available Software Packages

Many software packages include choropleth mapping tools. Table 1 shows selected software packages that include a choropleth mapping tool. Many of them have a platform for side-by-side multiple-map visualization.

**Table 1.** Software Packages containing a choropleth mapping tool and three components that each software provides. Component (1) is an automated method to create multiple choropleth maps with identical class intervals. Component (2) is a function that allows users to promptly recreate class intervals at any zoomed-in area. Component (3) is linking and brushing for synchronous exploration of multiple choropleth maps. [a] Tableau is an information visualization environment that includes a utility to draw multiple choropleth maps. [b] Vega-Lite is an open-source graphing library containing a choropleth mapping tool.

| | | Name of Software | Side-by-Side Visualiza-tion | (1) Identical Class Intervals | (2) Local Class Intervals | (3) Linking and Brushing |
|---|---|---|---|---|---|---|
| (A) Commercial Software Packages | Desktop-Based | ArcMap | ✔ | | | |
| | | ArcGIS Pro | ✔ | | | ✔ |
| | | Tableau Desktop | ✔ | ✔ | | ✔ |
| | | Maptitude | ✔ | | | |
| | Web-Based | ArcGIS Online | | | | |
| | | eSpatial | | | | |
| | | Mapbox | | | | |
| | | iMapBuilder | | | | |
| | | CARTO | | | | |
| | | MapBusinessOnline.com | | | | |
| | | Social Explorer | ✔ | ✔ | | ✔ |
| | | MAPLARGE | ✔ | ✔ | | ✔ |
| | | Tableau Online [a] | ✔ | ✔ | | ✔ |
| (b) Freely Available Software Packages | Desktop-Based | GeoDa | ✔ | | | |
| | | QGIS | ✔ | | | |
| | | STARS | ✔ | | | |
| | | Descartes | ✔ | ✔ | | ✔ |
| | | Cartographic Data Visualizer | ✔ | | ✔ | |
| | Web-Based | Vega- Lite [b] | ✔ | ✔ | | |

**Table 1.** *Cont.*

| | Name of Software | Side-by-Side Visualiza-tion | (1) Identical Class Intervals | (2) Local Class Intervals | (3) Linking and Brushing |
|---|---|---|---|---|---|
| | Cancer Atlas [21] | | | | |
| | US Health Map [22] | | | | |
| (C) Web-GIS Applications Visualizing Built-In Data Only | Census Data Mapper [23] | | | | |
| | Uber and Lyft in San Francisco [24] | | | | |
| | The Opportunity Atlas [25] | | | ✔ | |
| | Census Explorer [26] | | ✔ | | |
| | Neighborhoods Over Time [27] | | ✔ | | |

In many multiple-choropleth-map-visualization packages, users either need to pool their data and classify the whole space–time series or must adopt constant classification intervals to compare the distributional patterns of the multiple maps. Specifically, users often need to create choropleth maps showing (1) the change of spatial distribution of a particular race across time (e.g., Figure 1), (2) the comparison of spatial distribution of different races in a multiracial urban area (e.g., Figure S2), or (3) the income distribution dynamics of poverty over multiple times (e.g., Figure S1). For example, each set of top five maps and bottom five maps in Figure 1 represents the percentage of Asians and Pacific Islanders from 1970 to 2010. Figure 1's (1A) to (1E) have the different classification intervals created for each of five maps individually. In this case, since every map has different class intervals, it is hard to tell which areas have experienced an increase or decrease in the percentage of Asians and Pacific Islanders. To do so requires that a user first compare the values of the classification bins across the maps, and then consider the spatial pattern evolution. On the other hand, Figure 1's (2A) to (2E) have the same classification intervals which enable direct comparison among five different maps and allow us to see the spatiotemporal change over time.

In the situation mentioned above, however, the graphical user interface (GUI) of conventional GIS software packages, such as ArcMap and ArcGIS Pro, does not provide an easy way to create multiple maps with the same class intervals. For example, no matter what the classification method is, ArcGIS Pro users normally click buttons a few times to create class intervals for each map separately. As a result, every map has different class intervals, as in Figure 1's (1A) to (1E) maps. In contrast, it is not trivial to create one classification having eight intervals for all five different datasets in Figure 1's (2A) to (2E) maps. The amount of effort required by the user to achieve this result is substantial. There are no automated processes for this within a few clicks of a button. On the other hand, ArcGIS Pro users can write a Python script to compute the same class intervals for multiple choropleth maps and visualize them. However, Python scripting involves a steeper learning curve.

To overcome the limitation mentioned above, some of the Web-based GIS applications provide a function that users can employ to automatically set the same classification intervals over different views, prime examples being the Census Explorer [26] and Neighborhoods Over Time [27]. In these cases, the same classification intervals over multiple maps allow users to see the change of spatial distribution of a variable over time. Although they provide consistent class intervals for multiple choropleth maps across different points in time, they do not provide a function to set consistent class intervals over multiple choropleth maps displaying different variables. For example, these Web applications do not allow users to compare the percentage of whites, the percentage of blacks, and the percentage of Hispanics on the same scale (e.g., maps like Figure S2). Another limitation of these two Web applications is that users can visualize only the data that the application owners provide, but users cannot visualize their data. Social Explorer does not have any of the issues mentioned above, but users can only visualize up to two maps at a time.

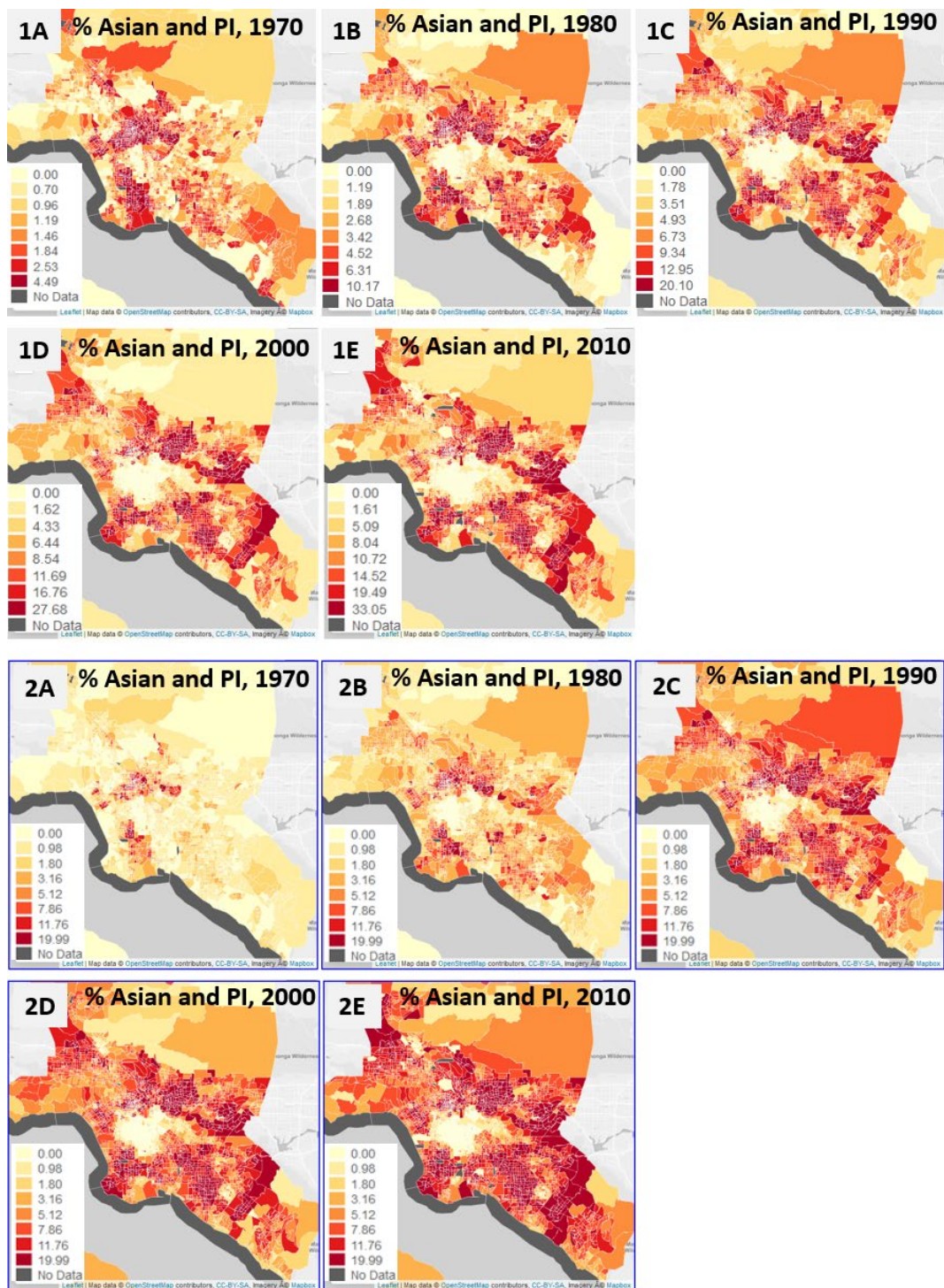

**Figure 1.** The distribution of the percentage of Asians and Pacific Islanders in 1970, 1980, 1990, 2000, and 2010, in Los Angeles. Maps (**1A**) to (**1E**) have different classification intervals, while maps (**2A**) to (**2E**) have the same classification intervals.

Another limitation of other popular GIS software packages, such as ArcGIS Pro and ArcMap, is that the class intervals are computed using all values within the study area. In the case of conventional choropleth mapping tools, if the user zooms in on the areas showing clusters of high or low values, in order to see the detailed distributional patterns within the zoomed-in area, the visualization will may show only limited variation. While this is understandable when considering a local area as one part

of the entire map, the user experience of zooming and panning map views is intended to focus on a specific area and examine the local structure of variability in the mapped attribute.

For example, Figure 2 shows what happens when users zoom in on a local region from the whole map. The left map shows the percentage of whites in the Los Angeles metropolitan area at the tract level. The right map shows the zoom-in area of the left map. After users zoom in on a small area such as the map on the right, all polygons have been assigned the same color. To reveal more variation in values in the right map, users would need to manually reselect the polygons, create a new layer with the selected polygons, and reassign the color scheme. The process requires several parameters to be re-tuned.

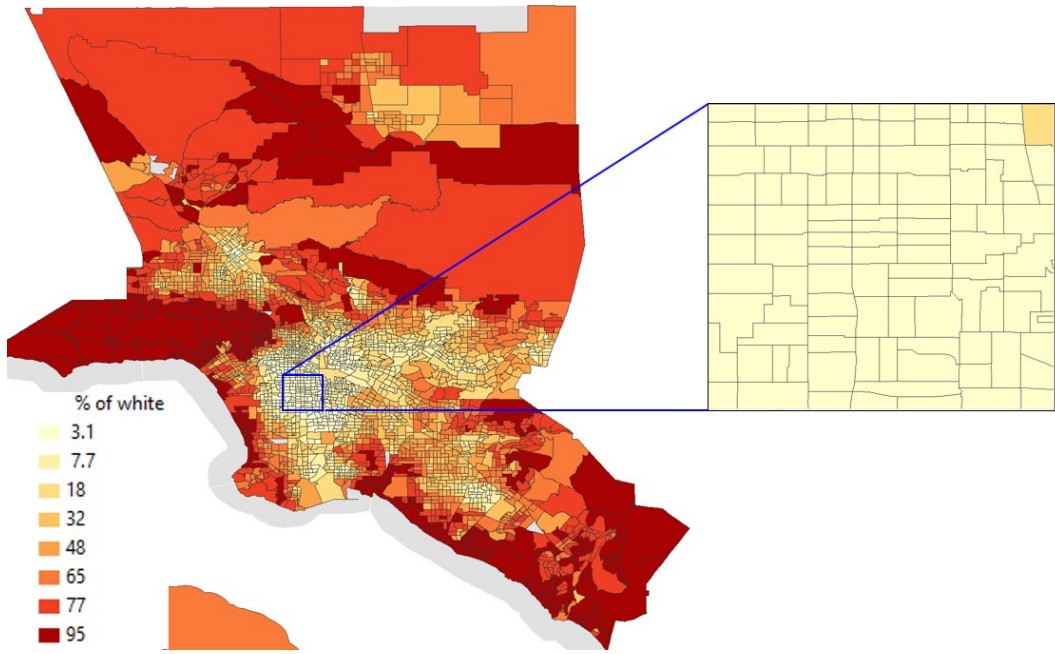

**Figure 2.** Limitation of Choropleth maps created by ArcMap and ArcGIS Pro.

*2.2. What is the Gap in Current Choropleth Visualization Implementation?*

Researchers have implemented three critical components for the effective visualization of multiple choropleth maps in various spatial extents since the 1990s (Table 1). However, existing software packages integrate only one or two of the three features but do not include everything in one choropleth mapping tool. For example, Descartes [28] and Social Explorer have the first and the third components, but not the second. On the other hand, CDV [29] has the second, but not the first and the third. The CDV facilitates multiple choropleth maps displayed side-by-side, but its platform does support the linking of multiple maps. The Opportunity Atlas [25] also has only the second component. On the other hand, Vega-Lite [30] is a JavaScript open-source visualization library for graphing that also contains a module for choropleth map visualization. The module has the first component, not the second or the third.

Furthermore, the innovative features of choropleth mapping tools cannot be easily combined with open-source libraries for Web application development, because many freely available GIS software packages are written in languages such as Java, C#, or C++ (see Section (b) of Table 1). These languages work best to make desktop applications. For example, freely available software packages such as Descartes [28] and CDV [29] are written in Java. The newer version of Descartes, known as V-analytics [31], is also written in Java. It is challenging to integrate these into existing Web-based GIS applications where visualization layers are written in JavaScript. For example, the implementation of the first and the third components of Descartes [28] cannot be integrated into Web applications written in JavaScript libraries such as Leaflet, D3, and PlotlyJS.

On the other hand, the commercial software packages, ArcGIS Pro, Social Explorer, and Tableau, support multiple choropleth-map visualizations with the first and third components in Table 1. However, they are not open-source software, nor are they distributed freely for re-use and modification. Thus, functionalities of those software packages are challenging to combine with open-source software packages such as Leaflet and PySAL [32]. MAPLARGE is the only one exceptional commercial software package that supports the first and the third components in Table 1; its source code is open, and it can be made compatible with open-source JavaScript libraries, such as Leaflet, OpenLayers, and ArcGIS API.

## 3. The Solution: Adaptive Choropleth Mapper

To implement open and Web-oriented multiple choropleth-map visualizations with all three components in Table 1, we developed the Adaptive Choropleth Mapper (ACM) with several open-source libraries: Leaflet (https://leafletjs.com/), D3 (https://d3js.org/), Geostats [33], Simple Statistics [34], Sweet Alert (https://sweetalert.js.org), and jQuery (https://jquery.com). ACM is a general mapping tool for the exploration of multiple choropleth maps. Users can enter their own data. ACM enables visualizing of up to 15 maps, side-by-side.

Other than the libraries mentioned above, the core part of ACM consists of one HTML file including the algorithm of ACM in JavaScript, one CSS file for styling, one configuration file, and two more files where the two inputs, variables, and geometry are saved separately. Users only need to change the data inside the two input files for variables and geometry to visualize the different datasets. The HTML file is the main program that users can run in their Web browser. The configuration file provides advanced options that users can change when they do not like the default settings.

Once ACM was developed, we created a Web GIS application called Longitudinal Neighborhood Explorer (LNE), where the Web-mapping tool, the ACM, was embedded. ACM is written in client-side scripting languages (i.e., JavaScript, HTML, and CSS). Thus, ACM can run on a local computer or be embedded in Web applications that have a client-server architecture. LNE is one example of the Web applications containing the ACM. While ACM allows users to visualize any data by replacing the two input files mentioned above, LNE was designed to visualize the Longitudinal Tract Data Base (LTDB) [35]. LTDB has spatiotemporal data representing race/ethnicity, socioeconomic, housing, age, and marital status at census tract level in the US. The LTDB was originally constructed from both the decennial census and American Community Survey data. Server-side programs in Python query the LTDB and pass the queried data to the client-side. Also, unlike ACM, LNE includes an interface that allows users to select variables and regions (Figure 3). The development of ACM and LNE are parts of The GeoSpatial Neighborhood Analysis Package (GEOSNAP) [36], a suite of spatial statistics and visualization tools for longitudinal neighborhood analysis.

### 3.1. Interface Design

The user interface of the Longitudinal Neighborhood Explorer (LNE) was designed to allow users to explore not only the temporal changes of distributional patterns of each socioeconomic and demographic variable, but also the difference in the distributional patterns of different variables at tract-level in different metropolitan areas or counties in the US. The user interface consists of two parts: the first allows the user to select the variables that they want to visualize, and the second is for visualizations of outputs in response to the user's selections, to create various views of maps. Figure 3 shows the user interface of the first part. Users have an option to select regions defined in terms of states, metros, or counties. Some of the data are not available in some years, especially in 1970. The unavailable data are represented by strike-out text. As users change the year at the top of the interface, they can check the availability of each variable over time. Users can select as many as variables they want by using the checkbox of each variable. Once the choice is submitted by clicking the button on the left bottom in Figure 3, the second part of the interface (e.g., Figure 4) appears. The second part is where ACM is embedded.

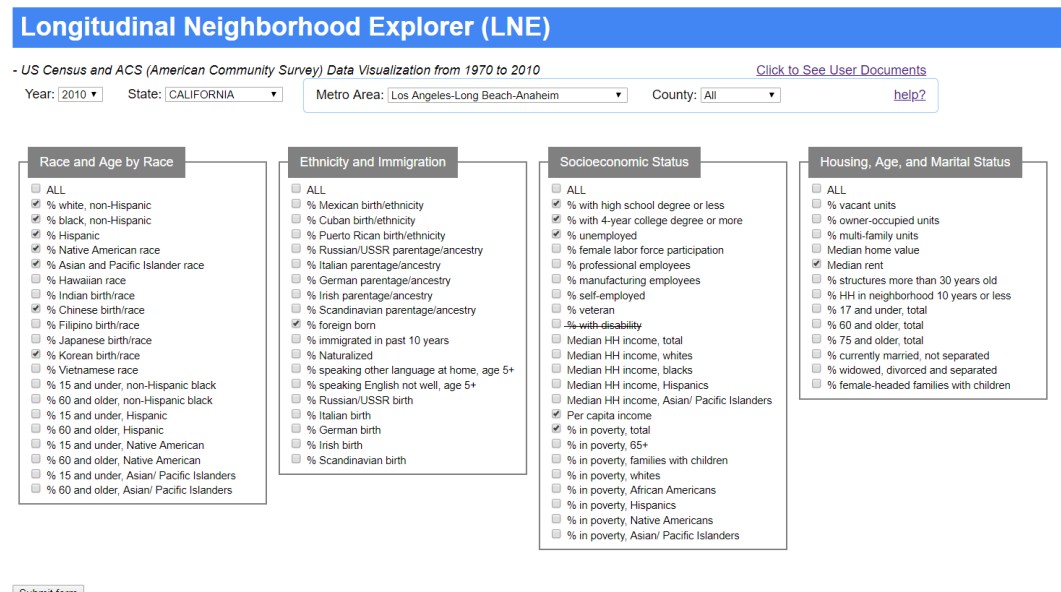

**Figure 3.** The first part of the user interface of Longitudinal Neighborhood Explorer (LNE). It was designed to get inputs from users. The LNE is available at http://sarasen.asuscomm.com/LNE/intro.html.

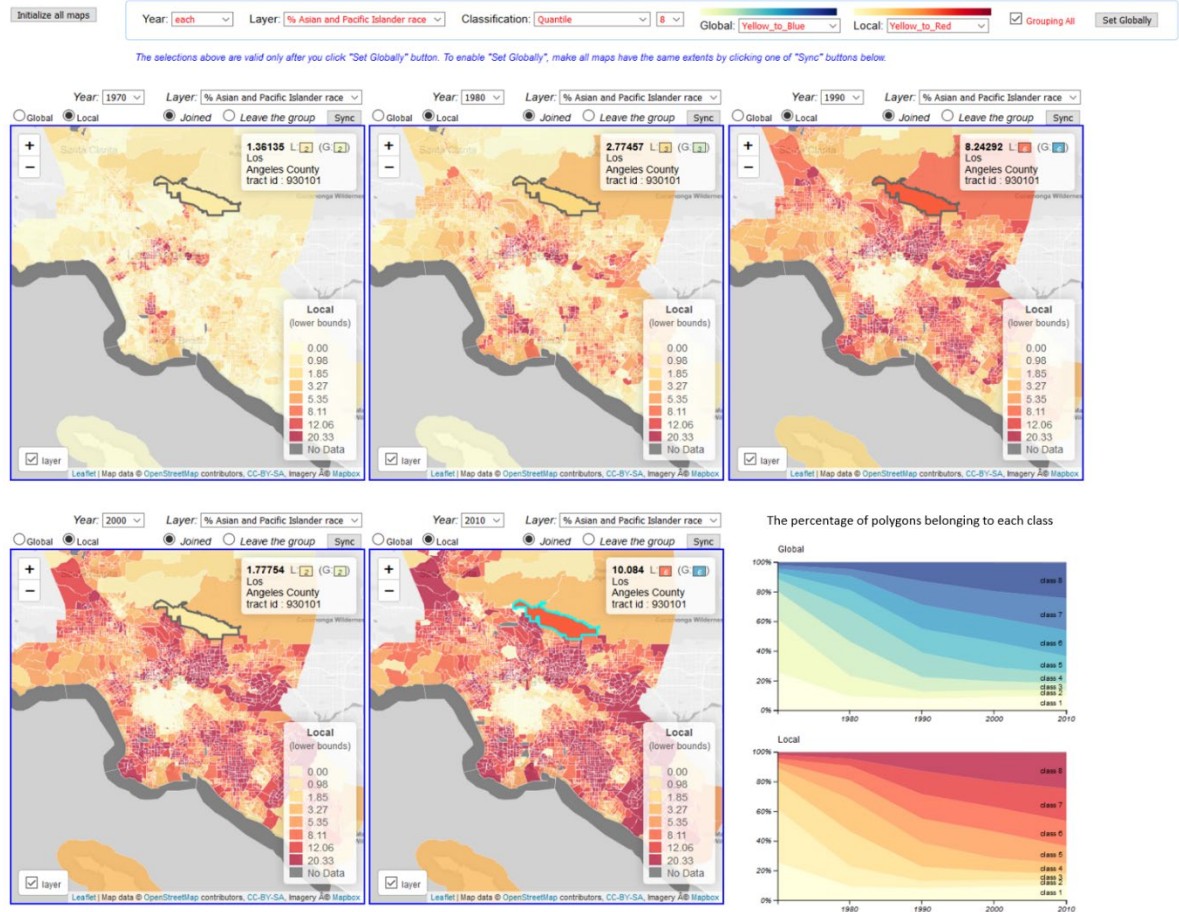

**Figure 4.** The second part of the user interface of Longitudinal Neighborhood Explorer (LNE). Maps represent the percentage of Asians and Pacific Islanders with the same classification intervals over all maps. It is one of the views created from the Adaptive Choropleth Mapper.

In the second part of the interface, users can visualize the selected variables, one by one, using ACM. For example, Figure 4 shows the second part of the user interface, where the selected variables show up in the "Layer" dropdown box. Figure 4 shows one of the views that can be created by ACM. In this case, the percentage of Asians and Pacific Islanders was selected among the thirteen variables previously selected in Figure 3. In addition to this view, numerous views can be created depending on users' selections in Figure 3.

In terms of the user-interface design of the second part, since LTDB data that we are using for the visualization have five different periods (1970, 1980, 1990, 2000, and 2010), our platform uses five maps together to visualize distributional changes over time. On top of the interface, the options in the light-blue outlined box change the visualization of all five maps together. The "Year" dropdown box has each year as an option, as well as an "each" option that visualizes years in sequential order. Users also have an option to visualize five maps of the same year. The "Layer" dropdown box allows users to change the variable of all five maps at once. The drop-down box "Classification" allows users to choose one of the classification methods (see Section 3.2.2. for details). "Grouping All" is described in Section 3.2.3. Once all options in the light-blue outlined box are selected on the top, users are required to click "Set Globally" button to see the changes on the maps. In addition, users can also change the view of each map separately by changing options on the top of each map. Figure 4 shows the "Year" and "Layer" drop-down boxes that are available on the top of each map. By using these drop-down boxes, users can change the year and the name of a layer in each map.

## 3.2. Functionalities of ACM

### 3.2.1. An Automated Process to Create Multiple Choropleth Maps with the Same Class Intervals

The Adaptive Choropleth Mapper (ACM) provides an automatic way to compute and set the same class intervals across different choropleth maps. Traditional choropleth mapping methods create class intervals separately for each map, which hinders comparisons between multiple maps. Especially when users have two or more different choropleth maps representing the same variable at different points in time, different class intervals in different maps prevent users from comparing the differences in maps over time. However, the ACM supports the traditional classifications (Figure 5) and also allows users to create the same class intervals for all maps, with one button click (Figure 4). The different class intervals for each map are computed with all values in each map. On the other hand, the same class interval for multiple maps is computed by using all values in all five maps. All computations deciding the class intervals are performed on-the-fly on the client-side. Figure 5 shows the percentage of Asians and Pacific Islanders, with different class intervals for each of five different maps for each year. Because these five maps have different class intervals, it is confusing to identify the spatiotemporal patterns in the percentage of Asians and Pacific Islanders over time. On the other hand, Figure 4 shows the distribution with the same class intervals. This allows users to identify regions where the percentage of Asians and Pacific Islanders has increased or decreased over time.

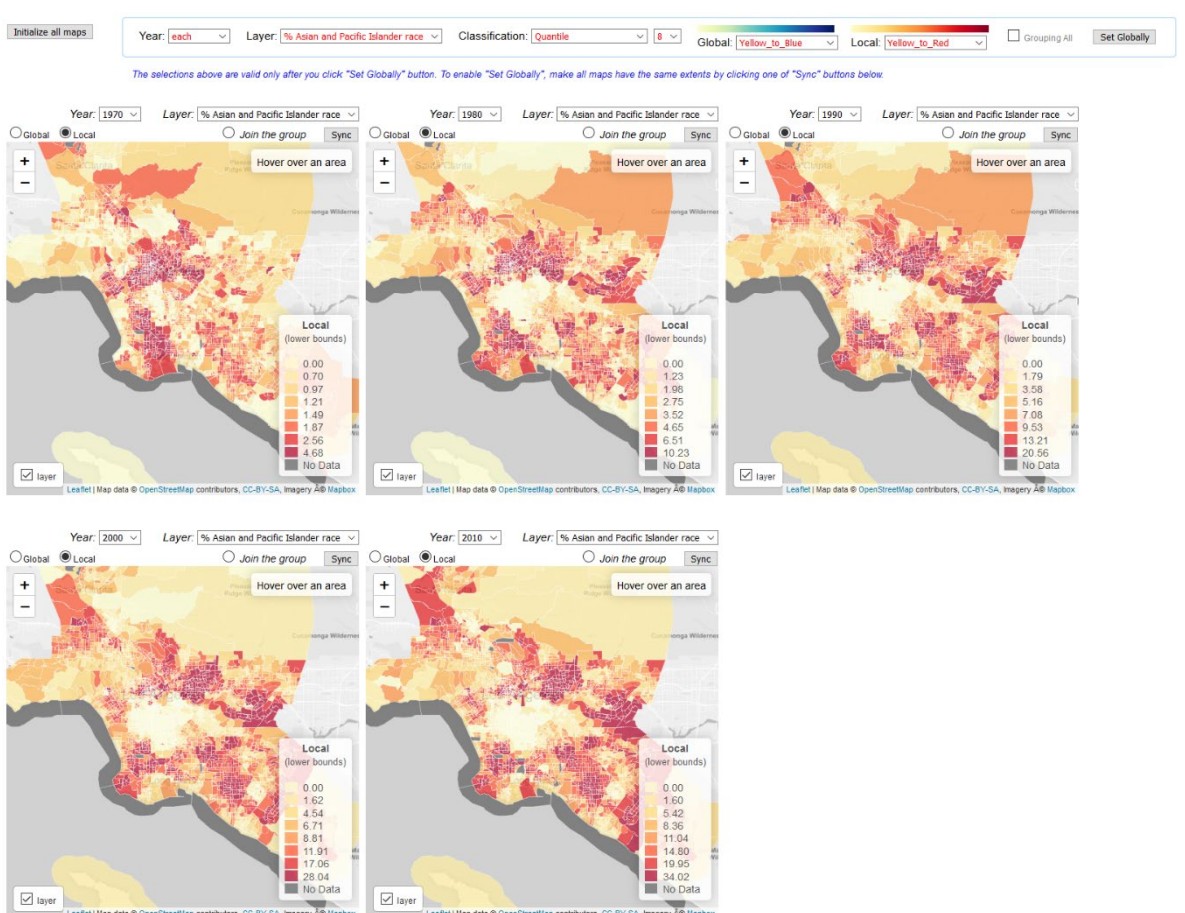

**Figure 5.** The percentage of Asians and Pacific Islanders with different class intervals of each map; it is one of the views created from the Adaptive Choropleth Mapper.

### 3.2.2. Class Intervals Adaptive to a Map Extent

The second function of Adaptive Choropleth Mapper (ACM) is paired visualization of choropleth maps, with both global and local classifications. ACM provides several classification methods: equal intervals, quantiles, standard deviation, arithmetic progression, geometric progression, and natural breaks. Because the Jenks natural breaks classification method is computing-intensive, we used the improved algorithm called CKmeans [37]. Jenks' and CKmeans produce the same result, but CKmeans has significantly improved computation time. CKmeans is available in Simple Statistics [34]. For the other classification methods, Geostats [33] was used.

For each classification method, the ACM provides users with an option to see maps with either a global classification or a local classification in any spatial extent. Global classification refers to the traditional choropleth mapping approach—i.e., one classification for all values in the entire study area. However, as we identified in Figure 2, the limitation of the global classification is that it gives some of the zoomed-in areas of the choropleth maps very limited visual variability; this may mask local spatial variation in the attribute under consideration. To overcome this limitation, ACM aims to retain the number of visually apparent classes in the zoomed-in areas. To achieve this, ACM provides local classification, as well as global classification. Local classification, which is a novel aspect of ACM, adaptively recomputes the class intervals by using values only within the current extent of the map. Whenever the user changes the map extent by panning and zooming, the classification intervals also change. In addition to the local/global classification, ACM provides an option to change the classification methods, the number of classes, and the color scheme in any spatial extent.

Figures 6–8 show how global and local classification work, using the percentage of Asian and Pacific Islanders as an example. In Figure 6, on the top of all maps, each yellow-to-blue and yellow-to-red

color scheme was assigned to global and local classification, respectively. At the top of each map, users have an option to select between local and global classifications. When the "Local" radio button is on, the map with local classification intervals is shown (e.g., Figure 6B). When the "Global" radio button is on, the map with global classification intervals is shown (e.g., Figure 6C). Figure 6B,C maps show the same area that is the zoomed-in area of Figure 6A. The Figure 6D map shows the same area as the maps of Figure 6B,C, but shows a base layer, which is OpenStreetMap. By unchecking the "Layer" at the bottom left corner of each map, users can see the base layer of the map.

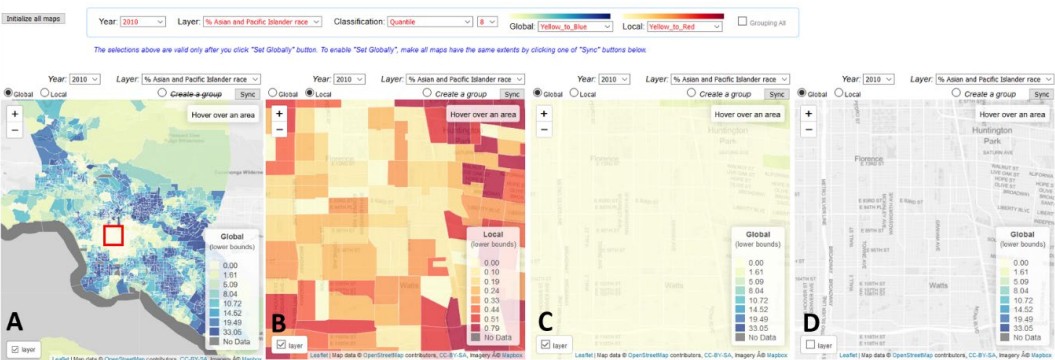

**Figure 6.** One of the views of ACM. Various views can be created by using options at the top of each map. Maps show the percentage of Asians and Pacific Islanders in the Los Angeles metropolitan area in 2010. The area within the little box at the center of the first map (**A**) is the same area of the other three maps (**B**–**D**). The "Local" radio button is clicked on map (**B**), while the "Global" radio button is clicked on the rest of it.

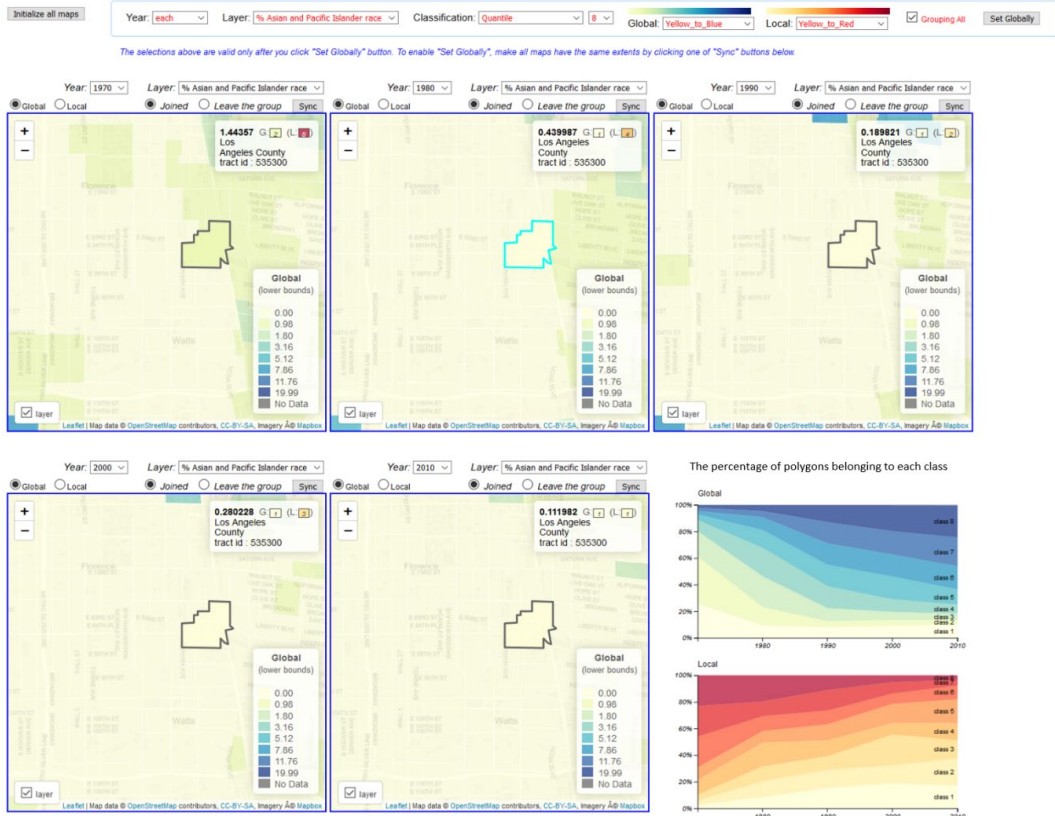

**Figure 7.** The percentage of Asians and Pacific Islanders in a zoomed-in area with global classification. The "Global" radio button is clicked at the top-left corner of each map. In this example, the quantile classification method was chosen. Figure S3 shows an example of how the maps look different in the same area, but with the natural break classification method.

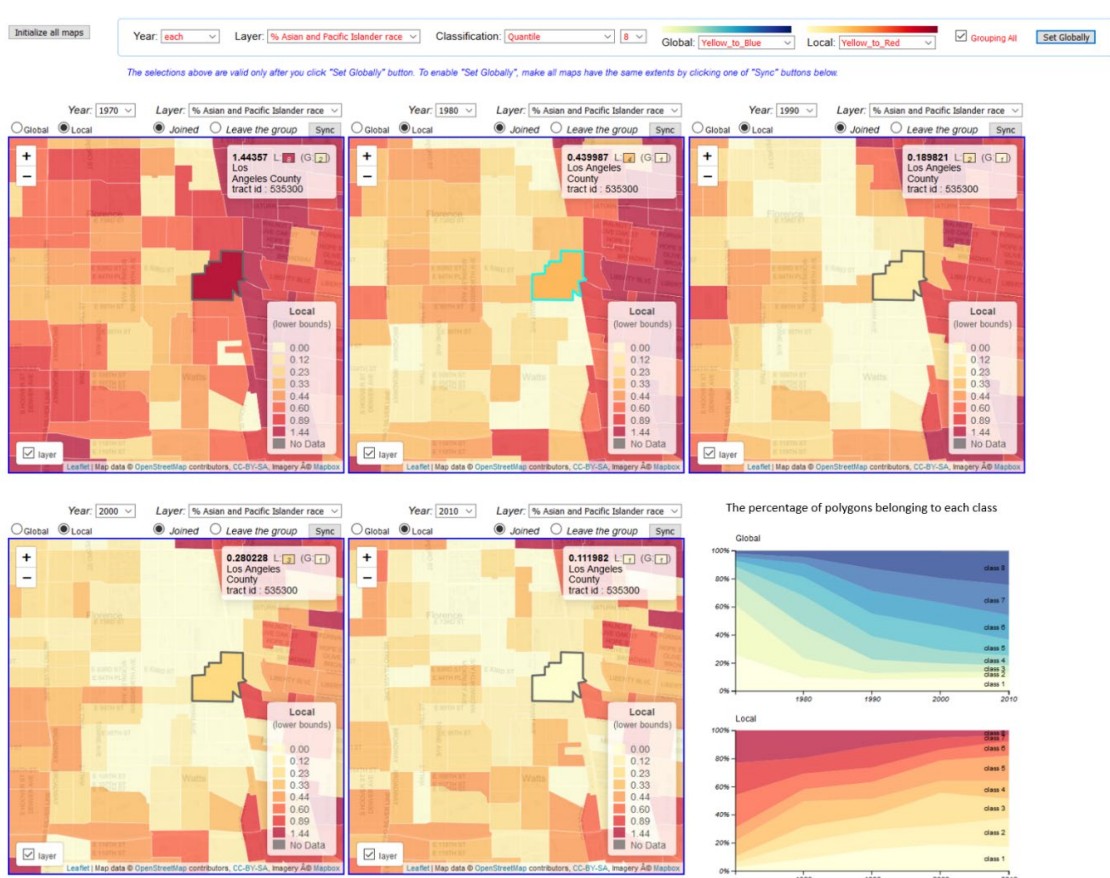

**Figure 8.** The percentage of Asians and Pacific Islanders in a zoomed-in area with local classification. The maps above show the same area as the maps in Figure 7. The "Local" checkbox is checked at the top-left of each map. In this example, the quantile classification method was chosen. Figure S4 shows an example of how the maps look different in the same area with the natural break classification method. The animated version with narrations of Figures 7 and 8 is available at http://sarasen.asuscomm.com/acm/Fig8.

The dual visualization of choropleth maps with both global and local classifications reveals changes in the distributional patterns across time in the zoomed-in areas. For example, maps in Figures 7 and 8 show a subregion of Figure 6A and also the same area as Figure 6B,C. Figure 7 shows the class intervals with global classification, which means that the class intervals are computed with values of all polygons within the entire Los Angeles metropolitan area. On the other hand, Figure 8 shows the class intervals of local classification, which means that class intervals are computed with values of the polygons within the current map extent. When Figures 7 and 8 are compared, Figure 8 shows much more variability in color lightness than Figure 7. Also, Figure 8 shows more distributional changes between class intervals over time than Figure 7. While traditional choropleth mapping tools support only views in Figure 7, ACM supports both views in Figures 7 and 8.

In terms of the map legend, Adaptive Choropleth Mapper (ACM) shows local or global classification intervals based on the user's selection (see the legend on the bottom-right corner on each map of Figures 7 and 8). The legend on the top-right corner of each map also shows the class number of a selected polygon (i.e., a highlighted polygon) in both global and local classifications. For example, on the second map in Figures 7 and 8, the highlighted polygon belongs to Class 4, using local classification, and Class 1, using global classification intervals. When users place their mouse over one of the polygons (tracts), the selected polygon is highlighted, and the information on the top-right corner legend changes.

### 3.2.3. Linking and Brushing for Exploration of Multiple Choropleth Maps

The third function of Adaptive Choropleth Mapper (ACM) is linking and brushing across multiple choropleth maps in terms of map extents and class intervals. G. Andrienko and N. Andrienko described the concept of linking multiple choropleth maps and their implementation in Descartes [28]. The linking of ACM was motivated by Descartes, but the ACM has some improvements—i.e., ACM uses linking for both full maps and partially selected maps (see Figure 9 for details). In addition to linking, ACM allows users to brush across a series of maps to highlight specific values/polygons.

The purpose of linking in ACM is to minimize the steps to make multiple maps have the same map extents and class intervals. This facilitates a direct comparison of multiple choropleth maps in any local area. On the interface of ACM, the buttons related to the linking are a "Grouping All" checkbox on the top of all maps, radio buttons such as "Create a group", "Join the group", "Joined", and "Leave the group" on the top of each map, and each "Sync" button on the top right of each map.

Linking in ACM has two modes: continuous linking and one-time linking. Continuous linking works in the following way: Once any of the maps joins the group, (1) the joined maps are outlined in blue to indicate the maps that will be linked, and (2) the maps in the group are adjusted to have the same map extents and class intervals whenever the user zooms and pans any of the maps. The three maps in Figure 9 show how the continuous linking works. On the interface, the user can only see three maps on each row, at one time. Three map views on each row show the status after each user operation. The blue outlines of the maps indicate the maps that are joined in the group. Arrows indicate the map that changes at each step. For example, Figure 9B1 changes from Figure 9A1 and changes to Figure 9C1. Figure 9 was created to show a step-by-step change. Only one map has a change at each step, and the changing map is represented in a cyan arrow. The Longitudinal Neighborhood Explorer has five maps. Among them, the three maps were captured for the demonstration in Figure 9.

In the example, a hypothetical demographer wants to compare the distributional patterns of different racial groups in a subset of Los Angeles' neighborhoods. At the initial stage, on the top of the interface, she picks the quantile classification as a classification method of the choropleth maps to be visualized, and then selects each of three different variables: percentage of whites, percentage of blacks, and percentage of Asians and Pacific Islanders on the top of each map (A1, A2, and A3 of Figure 9). At this stage, each map has a different set of classification intervals. Since she is first interested in examining the distribution of whites in the southwest area of Los Angeles County, she zooms in on the region on the leftmost map (Step 1). As a result of the first step, Figure 9A1 has changed to Figure 9B1. While she is examining the distribution of whites on Figure 9B1, she wants to compare the distribution of blacks and Asians and Pacific Islanders with the distribution of whites in the same area of Figure 9B1. For this comparison, she clicks the "Create a group" button on the top of Figure 9B1 (Step 2). After she clicks the button, the button changes to two buttons, "Joined" and "Leave the group", and the left map is outlined in blue as a result of the second step (Figure 9C1). On the next step, she also clicks the button "Join the group" on the top of the middle map, Figure 9C2 (Step 3). At this stage, the middle map view changes from Figure 9C2 to Figure 9D2. As a result, the map of Figure 9C2 has adjusted to have the same class intervals and map extent with the map of Figure 9C1. The Figure 9D1,D2 maps show after this adjustment is done. Now, she can have a direct comparison in the distribution of whites and blacks in the same region, with the same class interval in Figure 9D1,D2. Next, our demographer clicks the button "Join the group" on the top of Figure 9D3 (Step 4). At this stage, the right map changes from Figure 9D3 to Figure 9E3. Finally, Figure 9E1–E3 maps show the same area and represent the three different variables with the same class intervals, so she can compare the distribution of whites, blacks, and Asians and Pacific Islanders in the area of interest. She can also stop linking anytime by clicking "Leave the group" on the top of each map.

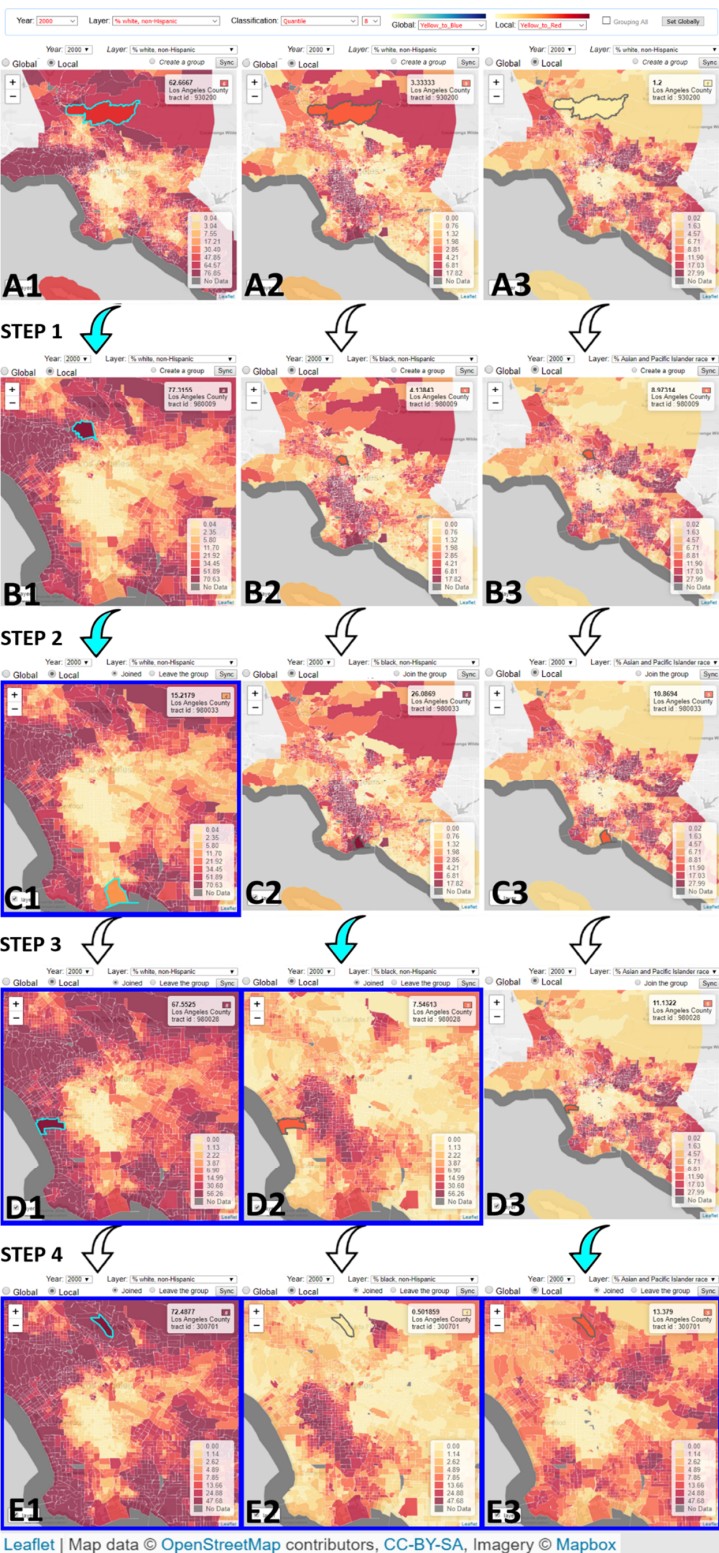

**Figure 9.** Continuous linking for a comparison of three variables: percentage of whites, blacks, and Asians and Pacific Islanders (from left to right). Step 1: The left map has been zoomed in on. Step 2: The "Create a group" button was clicked on the left map. Step 3: The "Join the group" button was clicked on the middle map. Step 4: The "Join the group" button was clicked on the right map. The animated version with narrations is available at http://sarasen.asuscomm.com/acm/Fig9.

Continuous linking provides several combinations in terms of grouping maps. For example, the maps in Figure 10, from left to right, represent (A) percentage of whites, (B) percentage of blacks, (C) percentage of Hispanics, (D) percentage of Chinese, and (E) percentage of Koreans. In the map views, only (D) and (E) are joined in a group (linked)—i.e., the two maps have the same map extent and the class intervals, so direct comparison between the two maps is possible. On the other hand, all the rest of it (Figure 10A–C) is not joined in a group (not linked), so each of the three maps can be explored individually with its own class interval and map extent. In this case, if the user wants to compare all five maps together, she can join the maps of Figure 10A–C in a group by clicking "Join the group" radio button on the top of each map. Figure 11 shows the views after all maps are joined. On the maps in Figure 11, users can have a uniform comparison of all maps.

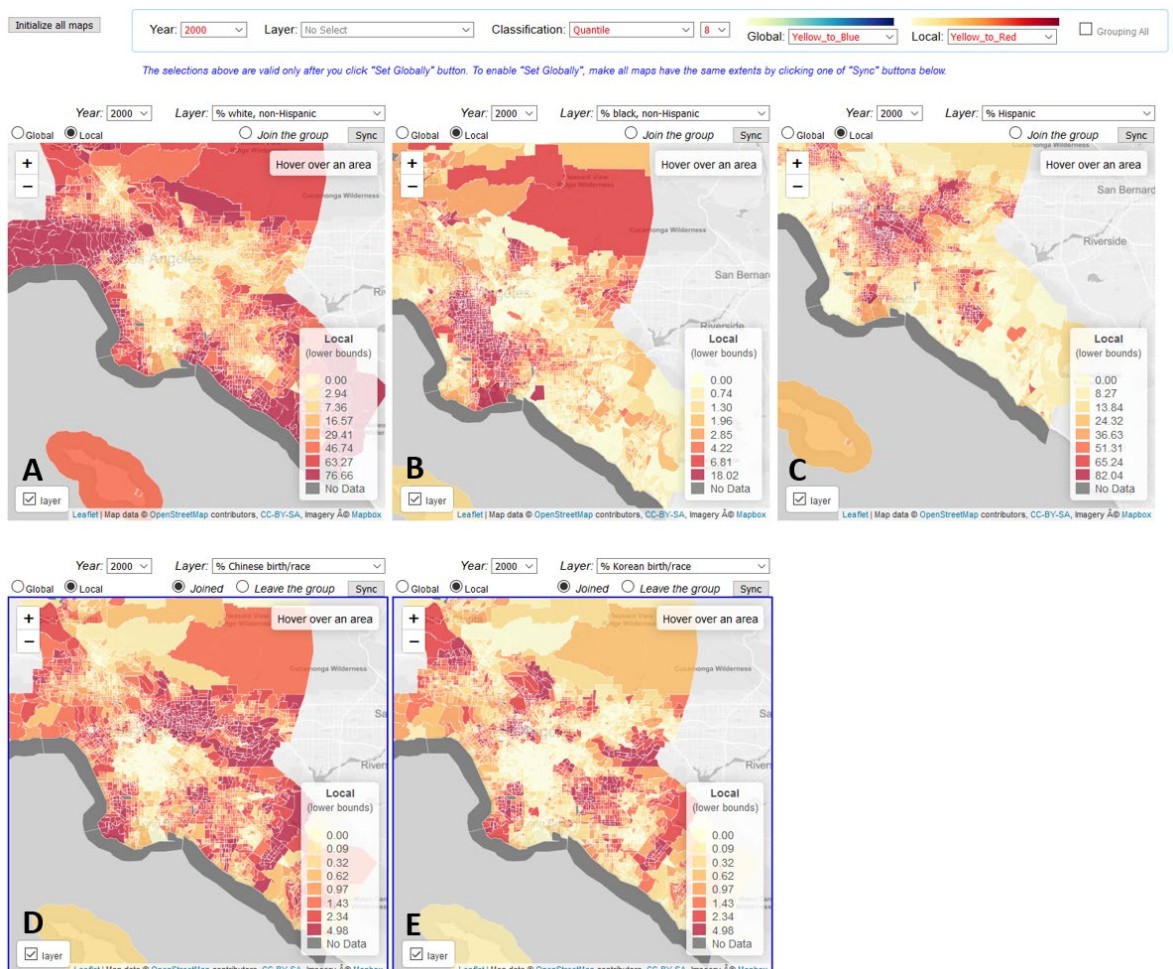

**Figure 10.** Linking partially selected maps. Maps (**D**) and (**E**) were joined in a group. The maps from left to right represent (**A**) percentage of whites, (**B**) percentage of blacks, (**C**) percentage of Hispanics, (**D**) percentage of Chinese, and (**E**) percentage of Koreans.

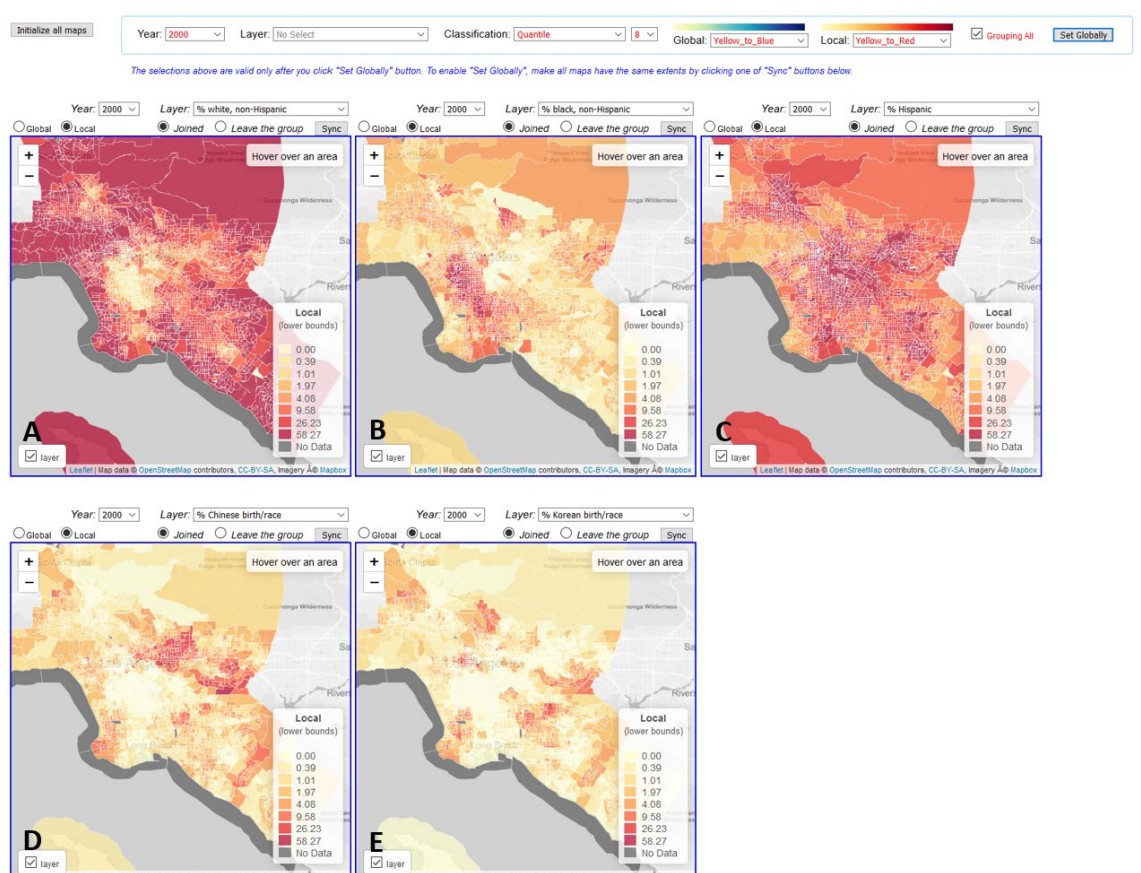

**Figure 11.** Linking all maps. The maps from left to right represent (**A**) percentage of whites, (**B**) percentage of blacks, (**C**) percentage of Hispanics, (**D**) percentage of Chinese, and (**E**) percentage of Koreans.

The different class intervals between Figures 10 and 11 show the necessity of linking selected maps. The class intervals in the maps of Figure 10D,E were computed with the values of polygons within the two selected (i.e., blue outlined) maps while the class intervals in Figure 11D,E were computed with values of polygons within all five maps. Therefore, the maps of Figure 10D,E result in a clearer comparison, with a bigger difference in color lightness than the maps of Figure 11D,E.

Furthermore, continuous linking allows users to quickly create a series of maps revealing the spatiotemporal change in any zoomed-in regions. For example, both Figures 4 and 8 show the percentage of Asians and Pacific Islanders in 1970, 1980, 1990, 2000, and 2010, from the left map to the right map. Figure 8 is a zoomed-in area of Figure 4. In this case, "Grouping All" was clicked at the initial stage so that all maps were joined in a group (linked) at once. By virtue of continuous linking, the maps in Figure 8 were created from Figure 4, after zooming and panning one of five maps. The video demo at the end of Figure 8's description shows how Figure 8 can be easily created from Figure 4. In other words, in ACM, the process of creating map views of Figure 8 from Figure 4 can be done with three steps: (1) check the "Grouping All" checkbox, (2) click the "Set Globally" button, and (3) zoom in on a subregion of their interest in any of five maps. In this third step, all maps are automatically adjusted to have the same map extent and class intervals. If each map in Figure 8 was created individually by using conventional GIS software other than ACM, it would have required more steps (see Section 2.1. for details).

The Adaptive Choropleth Mapper (ACM) also provides one-time linking. Users usually keep zooming and panning the map until they find a particular region that they are interested in. In this situation, the continuous linking of multiple maps is computationally expensive, as the program checks whether each polygon is within the map extent every time the user pans or zooms, and it

recomputes the class interval with values of polygons within the current map extent. We found that the use of continuous linking over 3000 polygons of five maps slows down the performance in moderately powerful computers. In this case, a "Sync" button can be used for the one-time linking of all maps. Figures 7–11 show that every map has a "Sync" button on the top-right corner of each map. Each "Sync" button allows users to adjust all other maps to have the same map extent and the same class intervals with the map whose "Sync" button is clicked. The "Sync" button provides only one-time linking in terms of the class interval and map extent only for all maps. After one-time linking, if the user changes the map extent in any of the maps, the shifted maps no longer have the same class interval and map extent as the rest of the maps.

### 3.2.4. Temporal Pattern Visualization

The fourth function of Adaptive Choropleth Mapper (ACM) is a stacked chart representing the temporal change of each class in choropleth maps over time (see the bottom-right image of Figure 4, Figure 7, and Figure 8). The top stacked chart represents the percentage of the number of polygons belonging to each of the global classification intervals. The bottom stacked chart represents the percentage of the number of polygons belonging to each of the local classification intervals. Both global and local stacked charts show up only if all five maps have the same class intervals and the same map extents. The global stacked chart never changes, because the global classification intervals remain the same regardless of map panning and zooming. However, the local stacked chart changes based on local class intervals because they are recomputed every time users pan or zoom all maps together.

In some cases, the trend of temporal dynamics is different between an entire area and a local region. For example, the global and local stacked charts at the bottom-right corner of Figure 8 show the different temporal dynamics in the distribution of Asians and Pacific Islanders between the global and local extent, i.e., the global stacked chart with a yellow-to-blue color scheme shows Asians and Pacific Islanders have increased in the whole Los Angeles metropolitan area (i.e., the area in Figure 6A); however, the local stacked chart with a yellow-to-red color scheme shows that Asians and Pacific Islanders have decreased in the area within the current map extent, which is a relatively small subregion around Florence, within LA (i.e., the area in Figure 6D).

## 4. Conclusions

The Adaptive Choropleth Mapper (ACM) has many improved functionalities compared to conventional choropleth mapping methods. It provides the following functionalities: (1) an automatic way to set the identical class intervals for multiple choropleth maps, (2) local classification intervals to reveal detailed variation in zoomed-in areas, (3) linking all or selected maps in terms of the map extent and the class intervals, and (4) a stacked chart for visualizing temporal dynamics. To the best of our knowledge, ACM is the first open-source online tool that displays multiple choropleths maps side by side and has all the functions mentioned above. In particular, the concept of adaptive class intervals for multiple-choropleth-map visualization (described in Section 3.2.2.) is a unique feature of ACM which has never been implemented alongside the linking described in Section 3.2.3. In short, the unique feature of ACM is that no matter what parts of maps that users zoom in on to examine, the local classification of ACM never creates choropleth maps with too few classes. In addition, when users navigate one of the selected maps by panning or zooming, all the other maps move synchronously, in order to show the same extent in every map. The unique features of ACM help users examine spatial distributions in detail, i.e., maps never lose detailed variation in spatial distribution no matter what areas are zoomed in on multiple choropleth maps.

Another advantage of the ACM is that it was made from open-source software packages. The source code of ACM is open and freely available, so it is extensible with any other open-source software. For example, there are hundreds of Leaflet plugins that can be combined with ACM. Many users have contributed to the developments of these plugins (https://leafletjs.com/plugins.html). In addition, open-source plotting libraries, such as D3 and PlotlyJS, can also be combined with ACM.

ACM can be used to visualize any data, such as mortality and disease data, as well as census data, to any geographic extents. The Adaptive Choropleth Mapper (ACM) is available as an open-source project at http://sarasen.asuscomm.com/ACM under an MIT license. Once downloaded, the ACM easily shows the user's geographic data. The ACM provides a template for a single choropleth map, as well as multiple maps (up to fifteen). A template for a single choropleth map was not designed for side-by-side comparison of the distributional patterns of multiple variables, but the template provides paired visualization of a choropleth map, with both the global and local classification intervals described in Section 3.2.2. For more than two maps, the ACM supports all functions described in Section 3.2. In terms of potential users of ACM, anyone who does not have programming experience can use the ACM by replacing the input data. GIS experts who have a programming background can easily add ACM to their own Web applications.

This paper described the advantage of adaptively changing class intervals on local areas of multiple maps. It remains unknown how well people can cope with changing class intervals of ACM in order to form a coherent understanding of local spatial variability in geographic data. In the next study, a survey will be conducted (1) to test if it is a cognitively challenging activity to understand patterns with adaptively changing class intervals on multiple maps and (2) to examine if people acquire some misconceptions about the spatial patterns of the datasets because of the automatically changing class intervals.

**Supplementary Materials:** The following are available online at http://www.mdpi.com/2220-9964/8/11/509/s1. Figure S1: Percentage of people in poverty in 1970, 1980, 1990, 2000, and 2010. The same class intervals over 5 maps allow us to see areas that have experienced the increase of the percentage of people in poverty across time. Figure S2: The fair comparison of the distribution of the percentage of whites (1), the percentage of blacks (2), and the percentage of Hispanics (3). Users need to have the same classification intervals over three different maps to have a fair comparison. The data are from LTDB. Figure S3: The percentage of Asians and Pacific Islanders in a zoom-in area with global classification. The "Global" radio button is clicked at the top-right corner of each map. The maps above show the same area with the maps in Figures 7 and 8, but the natural break classification method was chosen in this case. Figure S4: The percentage of Asians and Pacific Islanders in a zoom-in area with local classification. The "Local" radio button is clicked at the top-right corner of each map. The maps above show the same area with the maps in Figures 7 and 8, but the natural break classification method was chosen in this case.

**Author Contributions:** Conceptualization, Su Yeon Han, Sergio Rey, Elijah Knaap, Wei Kang, and Levi Wolf; data curation, Su Yeon Han; methodology, Su Yeon Han, Sergio Rey, Elijah Knaap, Wei Kang, and Levi Wolf; software, Su Yeon Han; validation, Su Yeon Han, Sergio Rey, Elijah Knaap, and Wei Kang; formal analysis, Su Yeon Han; investigation, Su Yeon Han; writing—original draft preparation, Su Yeon Han; writing—review and editing, Su Yeon Han, Sergio Rey, Elijah Knaap, Wei Kang, and Levi Wolf; visualization, Su Yeon Han; supervision, Su Yeon Han and Sergio Rey; project administration, Su Yeon Han and Sergio Ray; funding acquisition, Sergio Rey.

**Funding:** This material is based upon work supported by the U.S. National Science Foundation under a project [grant number 1733705], titled 'Neighborhoods in Space–Time Context'. Any opinions, findings, and conclusions or recommendations expressed in this material are those of the authors and do not necessarily reflect the views of the National Science Foundation.

**Conflicts of Interest:** The authors declare no conflict of interest.

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
