# Peer review of "Adaptive Choropleth Mapper: An Open-Source Web-Based Tool for Synchronous Exploration of Multiple Variables at Multiple Spatial Extents"

_ijgi, doi:10.3390/ijgi8110509_

Round 1

Reviewer 1 Report

Some items in the bibliography lacks some elements, such as the publication year (e.g. [12] and [14]. [26] for the access date).

I think sections 1 and 2 could be more concise and avoid some repetitions, but I guess it is a matter of personal preference and the current length is adequate.

The paper is focused on the motivation of the approach (sections 1 and 2) and on the software usage (section 3), while information about the implementation is scarce, with only a mention of the libraries used at the beginning of section 3. Additional information would be interesting.

L53 "it is not trivial to compute and set the same classification intervals for each of the choropleth maps in conventional GIS software packages such as ArcMap and ArcGIS Pro": however, it is simple to achieve this aim using GIS software which allows scripting. I would point out that experienced users with scripting skills can accomplish tasks (1) and (2) quickly.

L73 "[19])": runaway parenthesis?

L76 "the second component has rarely been available in modern GIS software": one can argue that the reason it has not been implemented is because it can be very confusing, as the association between colors and variable values changes with the zoom level. It can be useful to highlight local patterns, but not to compare values in different parts of a large map/dataset.

L162 Again, I think you should mention the possibility of scripting the creation of choropleth maps. This possibility can be beyond the skills of many users, but nevertheless exists.

L172 "a function that users can automatically set the same classification intervals." -> "a function that users can employ to automatically set the same classification intervals."

L223 LTDB -> Longitudinal Tract Data Base (LTDB), acronyms must be explained at their first use.

L283 " upto" - > " up to"

L331 There is a typo in the web application (http://sarasen.asuscomm.com/LNE/index.html), in the graph title "The percentage of polygons blelonging to each class" (belonging). This is visible in Fig. 4.

L530 Unless, of course the user has scripted the whole process.

Author Response

I have attached the file

Reviewer 2 Report

In the submitted paper the authors describe the programmed application which can be used for synchronous exploration of multiple variables at multiple spatial extents. The presented open-source web-based is innovative and suitable for Choropleth mapping.

The paper is clearly written, the results are clearly presented by many figures. I only recommend a minor revision of the References section. The publication year is missing in some references. For example references [2], [9], [12], etc.

Reviewer 3 Report

Review of ijgi-616319

This manuscript reports on the development of a web-based tool for interactively exploring a series of linked choropleth maps that may represent different time slices of one variable or multiple variables. There is enough novelty in the tool that it makes a contribution to the web mapping literature, particularly its flexibility and customisability, but the manuscript needs substantial work before it should be accepted.

I detail the things I think need improving before the manuscript should be accepted.

The manuscript is overly long and much of the information in it is repeated several times. It should be significantly condensed and more concisely reported. I think you could easily lose at least 5 pages by communicating more concisely and not repeating yourself. As but one example, the tool’s ‘critical features’ are listed at least five times. At most, once the abstract, once in the introduction and once in the conclusions should suffice. Your comparison of different existing tools (which is scattered throughout the manuscript but discussed in the most detail in 2.2) could also be significantly condensed and communicated through a table that compares all the relevant dimensions of the tools, with the discussion of the tools then referring to this table. Web maps and Web GIS are conflated in the manuscript. They are not one and the same. Here you are presenting a web mapping tool, not a Web GIS. This point should be addressed throughout the manuscript. Care should also be taken to properly characterise other tools that you compare your tool with. Some tools are referred to in curious ways. Tableau is most definitely not a cartography package (though it can produce maps) and neither is it a GIS package. It is an information visualization environment. Maps of qualitative areal data are not choropleth maps and should not be discussed as such. They are area-class maps. Choropleth maps are maps of ordinal or interval/ratio data. On that note, the authors should also use cartographic terminology when describing their maps. The colour variations in area-class maps relate to colour hue, while the colour variations in choropleth maps manipulate colour lightness. It would be useful if the authors used terminology from information visualization and interactive cartography to describe what they have implemented – this will enhance the manuscript’s contribution to our broader knowledge about interactive web maps. As one example, your synchronisation of class intervals, areal extents, and zoom levels is a form of linked displays. See: https://infovis-wiki.net/wiki/Linking_and_Brushing. It appears that you can also brush across the maps in a series to highlight specific values. It is not entirely correct to claim that GIS packages require the user to individually set classification breaks for each of the maps in a map series. It is quite possible to save class breaks in a layer’s style file and apply that style to a new map frame with different data. The specifics of how to do this can vary between software packages, but see here, for example: https://pro.arcgis.com/EN/PRO-APP/TOOL-REFERENCE/DATA-MANAGEMENT/apply-symbology-from-layer.htm. What your tool makes it easier to do and that is not, to my knowledge, possible in GIS packages, is to compute a sensible set of class breaks across multiple datasets. It is also incorrect to claim that commercial GIS packages are incompatible with Leaflet and some other open source libraries. In just one example, there is the Esri Leaflet API that allows mashups between ArcGIS generated content and Leaflet. I am sure if I spent time looking, I would discover plenty of other opportunities to combine commercial and open source functionality. While I understand your argument about the utility of global vs local class breaks, I wonder if you have considered how well actual people can cope with changing class breaks being shown using the same symbols and forming a coherent understanding of the spatial variability of a pattern. I think you need to at least address the fact that this is unknown (how well people can cope with this) and suggest it as future work that should be carried out to understand more fully the usability and utility of your tools. I would not be surprised if this is a very cognitively challenging activity and people acquire some misconceptions about the spatial patterning in the datasets because of this. Lines 133: I have no idea what the authors mean by a “comparative statistics” view. The whole manuscript needs a good proofread by a native speaker of English. The English language expression is of a largely reasonable standard but there are many small errors – too many for me to detail individually. I note that at least one of the authors is a native speaker of English.

Reviewer 4 Report

accept

Author Response

Thank you so much!

Round 2

Reviewer 1 Report

L183 "open source": the notation "open-source" is used in the rest of the paper.

L210 "The HTML file is the main program that users can run it in their web browser." -> "The HTML file is the main program that users can run in their web browser." (deleted "it").

Author Response

Thank you so much. The issues that you mentioned have been fixed.

Reviewer 2 Report

I recommend accepting the paper in present form.

Author Response

Thank you so much

Reviewer 3 Report

In general this is a much improved version of the manuscript and the authors have done a good job to respond to my critiques of their original submission. The value proposition is much clearer now that repetition has been removed and Table 1 and additional discussion of other tools have been added.

I found a few small typos and awkward sentences that could be fixed up before the paper is put online. I do not need to see the manuscript again.

Line 88: …which supports spatiotemporal census data…

Lines 124-135: I think the point that should be emphasised is not that ArcGIS Pro (for example) cannot do these other types of classification, but it cannot do them easily. The amount of effort required by the user to achieve this result is substantial and there are no automated processes for doing these computations with a few button clicks.

Line 146: …from census data over time.

Line 175: one choropleth mapping implementation.

Line 217: …example of a Web-application where the ACM…

Line 221: I don’t understand the sentence: THE LTDB was originally collected from both census and American Community Survey.

Do you mean:

The LTDB was originally constructed from both the decennial census and American Community Survey data.

Line 236: strikethrough text.

Lines 251-253: Should this read?

In terms of the user-interface design of the second part, since the LTDB data that we are using for the example visualization exist for five different time points (1970, 1980, 1990, 2000, 2010), our platform uses five maps together to visualise distributional changes over time.

Line 270: 3.2.1 Identical Class Intervals over Multiple Maps

Line 271: …can create no only identical class intervals across all …

Figures 4/5 and Figure 1 show essentially the same thing. Why not use Figures 4/5 and refer to them instead of showing pretty much the same thing twice?

Line 295: same result as..

Line 302: I think you mean variability not viability?

Line 313: switch between

Line 317 & 318: basemap

Line 327: shows much more detailed

Line 372: class intervals continuously whenever the user zooms and pans

Line 374: do you mean that the user can only see three maps on each row at one time?

Line 378: the user’s operation

Line 388: zooms in on

Line 401: with identical class intervals

Line 413: Loreans

Line 414-5: and class intervals, so a fair comparison…

Line 416: of the maps

Line 421: linking partially…

Line 464: In brief, continuous linking is…

Line 465: computers. One-time linking…

Line 496: too few classes shown in the view by computing…
